# Bioconversion, Pharmacokinetics, and Therapeutic Mechanisms of Ginsenoside Compound K and Its Analogues for Treating Metabolic Diseases

Md. Niaj Morshed [1], Reshmi Akter [2], Md. Rezaul Karim [1], Safia Iqbal [1], Se Chan Kang [1,*]
and Deok Chun Yang [2,*]

1 Department of Biopharmaceutical Biotechnology, College of Life Science, Kyung Hee University, Yongin-si 17104, Republic of Korea; niajmorshed96@khu.ac.kr (M.N.M.); rezaulshimul@khu.ac.kr (M.R.K.); safiadorin@khu.ac.kr (S.I.)
2 Graduate School of Biotechnology, College of Life Sciences, Kyung Hee University, Yongin-si 17104, Republic of Korea; reshmiakterbph57@gmail.com
* Correspondence: sckang@khu.ac.kr (S.C.K.); dcyang@khu.ac.kr (D.C.Y.)

**Abstract:** Rare ginsenoside compound K (CK) is an intestinal microbial metabolite with a low natural abundance that is primarily produced by physicochemical processing, side chain modification, or metabolic transformation in the gut. Moreover, CK exhibits potent biological activity compared to primary ginsenosides, which has raised concerns in the field of ginseng research and development, as well as ginsenoside-related dietary supplements and natural products. Ginsenosides Rb1, Rb2, and Rc are generally used as a substrate to generate CK via several bioconversion processes. Current research shows that CK has a wide range of pharmacological actions, including boosting osteogenesis, lipid and glucose metabolism, lipid oxidation, insulin resistance, and anti-inflammatory and anti-apoptosis properties. Further research on the bioavailability and toxicology of CK can advance its medicinal application. The purpose of this review is to lay the groundwork for future clinical studies and the development of CK as a therapy for metabolic disorders. Furthermore, the toxicology and pharmacology of CK are investigated as well in this review. The findings indicate that CK primarily modulates signaling pathways associated with *AMPK*, *SIRT1*, *PPARs*, *WNTs*, and *NF-kB*. It also demonstrates a positive therapeutic effect of CK on non-alcoholic fatty liver disease (NAFLD), obesity, hyperlipidemia, diabetes, and its complications, as well as osteoporosis. Additionally, the analogues of CK showed more bioavailability, less toxicity, and more efficacy against disease states. Enhancing bioavailability and regulating hazardous variables are crucial for its use in clinical trials.

**Keywords:** ginsenoside compound K; metabolic disease; obesity; NAFLD; diabetes; osteoporosis





## 1. Introduction

In recent years, metabolic disorders have become a global health concern because of their rapidly increasing prevalence [1]. Global human health is severely challenged by the increasing incidence of metabolic diseases, which include type 2 diabetes (T2D), obesity, non-alcoholic fatty liver disease (NAFLD), gout, osteoporosis, hypothyroidism, and hyperthyroidism [2]. The International Diabetes Federation (IDF) reports that 537 million individuals worldwide had diabetes in 2021, with more than 90% of cases being type 2 diabetes [3]. By 2045, this figure is predicted to rise to 783 million. In addition, obesity has emerged as the biggest global problem of concern for public health, with the incidence of overweight and obesity rising sharply in recent years. In 2016, there were over 650 million and over 1.9 billion adults with obesity or adults who were overweight worldwide, respectively, making up around 39% of the world's population [4]. According to estimations in 2019, the worldwide incidence of gout was 0.1–0.3% [5], and the prevalence of NAFLD was 29.62% in Asia [5]. These findings demonstrate that metabolic disorders

represent a substantial challenge in human society due to the associated high mortality and morbidity; thus, understanding the pathophysiology and therapies of metabolic diseases is critical.

The pharmacological properties of Korean ginseng (*Panax ginseng* Meyer), which has been revered as one of the most renowned traditional Chinese herbal medicines for over two hundred years, are largely attributed to its bioactive triterpenoid saponins. These triterpenoid saponins, called ginsenosides, are divided into three categories: panaxadiol (PPD), panaxatriol (PPT), and oleanic acid [6]. More than 218 ginsenosides have been determined from multiple parts of the ginseng plant (roots, flowers, berries, and leaves), and these byproducts have become popular for research [7]. However, ginsenoside compound K (CK) is one of the most significant among these ginsenosides due to its high uptake and absorption rate into the human gastrointestinal tract and, ultimately, the systemic circulation [8]. Compared to other ginsenosides, CK has superior membrane permeability and a lower molecular weight, which contribute to its increased bioavailability [9]. Major ginsenosides undergo the transition to produce CK, which is rarely present in natural ginseng. Human gut bacteria and endophytes have been reported to use deglycosylated processes to bio-convert CK products. In the past year, yeasts have been metabolically altered by enzymes has become a viable alternative for producing CK [10]. According to current research, CK possesses pharmacological properties that include hepatoprotective, anti-inflammatory, anti-carcinogenic, anti-diabetic, anti-allergic, neuro-protective, and anti-aging activities [11].

CK is an active molecule that can reduce blood lipids and control glucose consumption [12]. Notably, CK is a regulator of *PPARγ* [13] and *AMPK* [14]. It has been demonstrated that AMPK increases glucose utilization, mobilizes lipid storage, and promotes autophagy to turnover macromolecular routes, so promoting the breakdown of biomolecules for the creation of energy [15]. Additionally, *PPARγ* influences lipid metabolism, which enhances sensitivity to insulin and glucose metabolism [16]. *AMPK* and *PPARγ* are the fundamental targets in metabolic disorders, including NAFLD, diabetes, osteoporosis, and obesity [17,18]. In addition, CK downregulates *PPAR*, *leptin*, *aP2*, and *C/EBP* adipogenic markers, which cause obesity, diabetes, and other metabolic diseases [19]. Moreover, the deregulation of metabolic processes associated with *TP53* results in various human pathologies, such as obesity, diabetes, liver, and cardiovascular illnesses [20]. CK significantly regulates the *TP53* expression in different disease states [21].

As a result, it is hypothesized that CK may be involved in metabolic illnesses by controlling inflammation and energy. However, due to the shortage of adequate information on CK regarding its use for the cure of metabolic illness, cytotoxicity is well documented, which prevents further development of the drug. This study includes an in-depth assessment of the use of CK to treat metabolic illnesses and the signaling pathways involved, as well as an analysis of its usual negative effects and pharmacokinetics. To give direction and evidence for CK research on metabolic conditions, we investigated the Google Scholar, Web of Science, PubMed, and CNKI databases up to December 2023.

## 2. Physical and Chemical Properties

CK (20-*O*-β-D-glucopyranosyl-20(*S*)-protopanaxadiol) is a minor tetra-cyclic triterpenoid that is rarely found in natural ginseng. Several approaches for CK synthesis have been described in detail (Section 4). CK is a white crystalline powder with a molecular weight of 622.9 g/mol, molecular formula $C_{36}H_{62}O_8$ [22], and PubChem CID 9852086. The physical and chemical properties of CK are shown in Table 1 (data collected from https://www.chemicalbook.com/ (accessed on 20 December 2023)).

**Table 1.** Physical and chemical properties of CK.

| Name | Compound K |
| --- | --- |
| Alias | 20-*O*-β-D-glucopyranosyl-20(*S*)-protopanaxadiol |
| CAS number | 39262-14-1 |
| Pubchem CID | 9852086 |
| Compound type | tetra-cyclic tri-terpenoid |
| Molecular formula | $C_{36}H_{62}O_8$ |
| Molecular weight | 622.9 g/mol |
| Form | powder |
| color | White |
| Solubility | DMF: 10 mg/mL; DMSO: 10 mg/mL; DMSO: PBS (pH 7.2) (1:1): 0.5 mg/mL |
| Density | 1.19 |
| pka | 12.94 ± 0.70 (Predicted) |
| Melting point | 181~183 °C |
| Boiling point | 723.1 ± 60.0 °C |
| LogP | 5.500 |
| Stability | Hygroscopic |

## 3. Pharmacokinetics, Safety, and Toxicological Studies of CK and Its Derivatives

The bioavailability rate of ginsenosides without conversion or modification indicates limited intestinal absorption [23]. According to research, some microorganisms and gut bacteria or soil fungi around ginseng roots hydrolyze ginsenosides to produce CK. It is essential to research the metabolic processes that control intestinal microbiota since it plays a crucial role in the biological transformation and therapeutic effects of CK [23]. Based on recent research on human metabolism, a high-fat diet greatly speeds up and increases the digestion of CK, and women have higher concentrations of CK than men [24]. After delivering Korean ginseng extract to ten healthy males for 36 h, the drug levels in their blood samples were reported in additional pharmacokinetic investigation on CK [25]. The mean time to achieve the Cmax (Tmax) of CK was greater compared to Rb1 (12.20 ± 1.81 vs. 8.70 ± 2.63 h), and the average highest plasma concentration (Cmax) of CK was substantially greater than the mean concentration of Rb1 (8.35 ± 3.19 vs. 3.94 ± 1.97 ng/mL). Intestinal microflora probably converts Rb1 to CK because of the delay in CK absorption. Compared to Rb1, CK had a plasma half-life ($t_{1/2}$) that was seven times shorter. The findings of this study suggest that there is a notable distinction in the pharmacokinetics of CK and Rb1. In a different study, 76 participants were given either a placebo or CK in seven individual doses taken orally (25, 50, 100, 200, 400, 600, and 800 mg) while they were fasting; the exposure to CK grew linearly between 100 and 400 mg, and the time range to attain Tmax was 1.5–6.0 h. After the seventh administration, the steady state was reached, and there were no serious adverse effects (AEs) reported. Watery stool (diarrhea) and stomachache were the most commonly reported AEs. All AEs were mild to severe, and the majority of them were cured quickly without any intervention. These findings demonstrated that CK was both safe and well tolerated for the course of the treatment [24,26].

According to toxicity tests, CK was applied on 3T3L1 pre-adipocyte cells in a dose-dependent manner. The maximum concentration of 40 μM did not affect the viability of the cells [27]. Fang et al. examined ginsenoside CK for cytotoxicity at various concentrations (0.2–10.0 μM). They observed that CK concentrations below 10 μM had no discernible impact on the survival rate of HaCaT keratinocyte cells [28]. The osteoblastic cell line MC3T3-E1 was exposed to CK at various concentrations (0.1–10 μM) and did not exhibit any appreciable toxicity [8]. When ginsenoside CK was evaluated on HepG2 cells, substantial cytotoxic effects were detected with increasing concentrations of ginsenoside CK up to 30 M compared to the control group [29]. Additionally, different CK doses (5–40 μM) were applied to HepG2 cells for 24 h to assess the cellular toxicity of CK. Even at dosages of 40 μM, CK did not exhibit any cellular damage [30]. However, the administration of a higher dose of CK (10 μM) significantly boosted the development of HT22 hippocampus

neuron cells [31]. After the treatment with 1.25–10 µM of CK, the rate of survival of the L02 cells climbed dramatically. CK showed no toxicity at any test concentration on L02 cells [32]. Human fibroblast-like synoviocytes RA-FLS cells and murine macrophage cells were treated at the same concentrations of 0.1–5 µM for the treatment of rheumatoid arthritis. The results showed that these cells were not affected by CK at a concentration of ≤5 µM [33]. Gu et al. evaluated the possible cytotoxicity of CK on MIN6 mouse pancreatic β-cell at various doses (2–32 µM). CK showed minimal effect at 16 µM and decreased the cell viability at 32 µM [34]. CK has a time- and concentration-dependent mild to moderate cytotoxic impact on cancer cells. The most susceptible to CK exposure were Hk-1 cells (a cell line used to study nasopharyngeal carcinoma), as 41.1–88.0% of cell mortality was seen at low levels (10–20 µM) [35]. Boopathi et al. reported that CK exhibits negligible cytotoxic effect on A549 lung cancer cells, Caco-2 colorectal cancer cells, and MCF-7 breast cancer cells at 12.5 µg/mL, whereas normal cell Raw 264.7 demonstrated less toxicity at 6.25 µg/mL [36]. At concentrations ranging from 8 to 64 µmol/L, compound K inhibited the growth of HT-29 cells in a dose-dependent manner; the dosage that produced 50% inhibition of growth (IC50) was 32 µmol/L [37]. Oral CK delivery to rats and mice in a toxicity trial did not result in toxicity or death at the maximal doses of 8 and 10 g/kg, respectively [38]. During a beagle toxicity investigation, dogs in the 36 mg/kg group experienced considerable weight loss and reversible hepatotoxicity. There was no discernible toxicity in the animals in the 4 and 12 mg/kg groups [39]. Table 2 elucidated the cytotoxicity of CK on different cell lines.

The bioavailability and solubility of CK and its derivatives have been enhanced via the application of modifications of structure and the use of nanocarriers. Igami et al. performed research to increase the solubility, dissolution rate, and bioavailability of CK by forming a complex with γ-cyclodextrin and improving oral bioavailability and water solubility [40]. In a different study, ginsenoside CK was used in conjunction with d-alpha-tocopheryl polyethylene glycol (PEG) 1000 succinate-iposomes to enhance solubility, target tumor cells, and reduce efflux [41]. In a different investigation, PCL (polycaprolactone), PEG, and TPGS CK-micelles (CK-M) showed better bioactivities and solubility [42]. When water solubility of CK-NPs/bovine serum albumin (BSA) and CK were evaluated, it was observed that BSA increased the water solubility of CK. BSA is a desirable carrier molecule due to its high biocompatibility, dispersive nature, and ability to conjugate with various target molecules [43]. In another investigation, the efficacy of medication loading was assessed when CK was loaded onto gold (G) NPs made with probiotic microbes (*Lactobacillus kimchicus DCY51T*) [6]. A recent study showed that probiotic (*Bifidobacterium animalis*)-mediated gold NPS of CK increases the capability of drug deliveries, biocompatibility, and oral bioavailability [44].

It has been demonstrated that CK is safe and well tolerated in both human and animal subjects. These preclinical findings imply that CK may be harmful to the liver. Although the relative weight of the kidney was high, there was no histological change, but nephrotoxicity should be noted. Abdominal pain and diarrhea were CK-related AEs observed in clinical trials. Both clinical trials and data on AEs associated with CK are scarce. Thus, more research is required to determine the processes underlying CK-induced gastrointestinal tract irritation and CK-induced damage, particularly hepatotoxicity. We studied ADMET (Absorption, Distribution, Metabolism, Excretion and Toxicity) analysis of CK by using the SwissADME (http://www.swissadme.ch/index.php, (accessed on 25 December 2023)), ADMETlab2.0 (https://admetmesh.scbdd.com/, (accessed on 25 December 2023)), and pkCSM (https://biosig.lab.uq.edu.au/pkcsm/, (accessed on 25 December 2023)) web servers, while the predicted toxicology properties were analyzed using the Protox-II webserver (https://tox-new.charite.de/protox_II/, (accessed on 25 December 2023)), where the drug likeliness and toxicity of CK was mentioned (Table 3).

**Table 2.** Cytotoxic studies of CK in several cell lines.

| Study Model | Concentrations | Method of Detection | Duration of Experiment | Result | Ref. |
|---|---|---|---|---|---|
| 3T3-L1 preadipocyte cell lines | (0, 10, 20, 30, and 40 μM) | MTS assay | 24 h | A high dose of CK (40 μM) did not affect cell viability. | [27] |
| HaCaT keratinocytes cells | 0.2, 0.4, 0.6, 0.8, 1.0, and 10.0 μM | MTT assay | 24 h | Below 10 μM showed safe survival of HaCaT cells. | [28] |
| MC3T3-E1 osteoblastic cell line | 0.01, 0.1, 1, and 10 μM | MTT assay | 48 h | No cytotoxicity was observed. | [8] |
| HepG2 | 1, 2, 5, 10, 15, 20, and 30 μM | MTT assay | 24 h | Cell viability reduces with increasing concentrations. | [29] |
| HepG2 | 5−40 μM | CellTiter 96 AQueous One Solution Cell Proliferation Assay kit | 24 h | CK did not exhibit any cellular toxicity below 40 μM. | [30] |
| HT22 mouse hippocampal neuron cell | 2.5, 5, and 10 μM | MTT assay | 24 h | CK can increase the survival of HT22 cells. | [31] |
| L02 Human liver cell line | 0.625, 1.25, 2.5, 5, 10, and 20 μM | MTT assay | 24 h | The cell viability appeared at the dosages of 1.25–10 μM. | [32] |
| RA-FLS and Raw 264.7 | 0.1, 0.5, 2.5, and 5 μM | MTT assay | 48 h | The survival rate of both cells was not impacted at doses of ≤5 μM. | [33] |
| MIN6 cell line | 2, 4, 8, 16, and 32 μM | MTT assay | 24 h | At 16 μM CK showed little toxicity on MIN6 cell | [34] |
| Hk-1 Nasopharyngeal Carcinoma cells, | 1–20 μM | MTT assay | 24 h | The IC50 of CK was 11.5 on HK-1 cells | [35] |
| A549 lung cancer cells, MCF7 breast cancer cells, Caco-2 human colorectal adenocarcinoma cells, and normal RAW 264.7 cells | 0, 3.125, 6.25, 12.5, and 25 μg/mL | MTT assay | 24 h | At 12.5 μg/mL concentration, CK showed considerable cytotoxic effect on A549 cells, MCF-7 cells, and Caco-2 cells growth. However, at 6.25 μg/mL, Raw 264.7 cells showed less toxicity. | [36] |
| HT-29 Human colon cancer cells | 8, 16, 32, and 64 μmol/L | MTT assay | 24 h | CK inhibited the growth of HT-29 cells in a dose-dependent manner. | [37] |
| HL-60 human myeloid leukemia cell line | 10, 20, 30, and 50 μM | MTT assay | 72–96 h | 24.3 μM was needed to achieve 50% growth inhibition (IC50) at 96 h. | [45] |
| U937, Jurkat, CEM-CM3, Molt4, and H9 leukemia cell lines | Did not mention | MTT assay | 96 h | The IC50 values of CK were as follows: 20 μg/mL for U937, 26 μg/mL for Jurkat, 36 μg/mL for CEM-CM3, 44 μg/mL for Molt 4, and 64 μg/mL for H9. | [46] |
| Rat and mice | 8 and 10 mg/kg, respectively | Acute oral repeated dose | 26 weeks | No indications of clinical harm or death after 14 days. A few variations were observed in this shift at weeks 9, 10, 12, 15, 17, 21–24, and 26. As a result, CK had a minimally negative impact on the animal's body weight. | [38] |
| Beagle dogs | 4, 12, or 36 mg/kg | Oral doses | 26 weeks | No obvious toxicity was shown by the animals in the 4 and 12 mg/kg groups. The 36 mg/kg group showed elevated plasma enzyme levels, localized liver necrosis, and a decrease in body weight. | [47] |

**Table 3.** ADMET properties analysis of CK.

| Physicochemical Properties | Compound K | Standard Range |
|---|---|---|
| Molecular weight (g/mol) | 622.9 | <500 |
| Num. rotatable bonds | 7 | |
| Num. H-bond acceptors | 8 | $\leq 10$ |
| Hydrogen bond donor | 6 | $\leq 5$ |
| Molar Refractivity | 172.26 | 40–130 |
| TPSA ($\text{Å}^2$) | 139.84 | <140 $\text{Å}^2$ |
| Lipinski | Yes; 2 violations | |
| Bioavailability Score | 0.17 | >0.1 |
| ADME | | |
| Human Intestinal Absorption | 54.344 | |
| GI absorption | Low | |
| Blood–Brain Barrier Permeability | $-1.038$ | 0–1 |
| Volume distribution | 1.061 | 0.04–20 L/Kg |
| Plasma–protein binding | 93.57% | <90% |
| Total Clearance (log ml/min/kg) | 0.46 | |
| CYP1A2 inhibitor | No | 0–1 |
| CYP2C19 inhibitor | No | 0–1 |
| CYP2C9 inhibitor | No | 0–1 |
| CYP2D6 inhibitor | No | 0–1 |
| CYP3A4 inhibitor | No | 0–1 |
| Toxicity | | |
| Hepatotoxicity | Active (0.69) | 0–1 |
| Carcinogenesis | Inactive (0.62) | 0–1 |
| Immunotoxicity | Active (0.96) | 0–1 |
| Mutagenicity | Inactive (0.97) | 0–1 |
| Cytotoxicity | Inactive (0.93) | 0–1 |
| Mitochondrial Membrane Potential | Inactive (0.70) | 0–1 |

## 4. Biotransformation of CK

It has been demonstrated that the naturally occurring ginseng plant does not provide significant amounts of the minor ginsenoside CK. Therefore, several studies have concentrated on using various techniques, including hydrolysis, enzymatic biotransformation, microbial transformation, etc., to convert major ginsenosides to CK. Additionally, endophyte biotransformation is an effective technique within microbial transformation because of its low cost, excellent selectivity, accuracy, and environmental safety [10]. Chemical hydrolysis conditions led to the nonspecific cleavage of glycone moieties at position 20, which in turn caused side reactions of hydroxylation, hydration, and epimerization. Furthermore, these methods proved to increase environmental contamination [48,49]. In contrast, owing to their notable selectivity, mild reaction conditions, and environmental compatibility, enzymatic or microbial conversion modalities have emerged as the most popular ones. Figure 1 demonstrates the biotransformation of major ginsenosides to CK via several pathways using enzymatic and microbial conversion methods.

### 4.1. Enzymatically Synthesis

One possible approach could be highly region-specific enzymatic transformation. Enzymatic techniques have been employed to produce CK from ginseng root extract, employing β-glucosidase (β-glu), β-glycosidase, and bi-composites of β-glucosidase.

The bioconversion of ginsenoside Rb1 to CK via β-glu is an effective production method in industry [50]. Microbes isolated from the soils of ginseng farms, soybeans, tea, the gastrointestinal tract of humans, kimchi, and other fermented items can be a source of β-glu [51]. Heat-resistant β-glu yields the highest amount of CK from protopanaxadiol-type ginsenosides [52]. Qin et al. used chromatography to purify a new ginsenoside-hydrolyzing β-glu from *Paecilomyces bainier* sp. 229 to increase the conversion rate of Rb1 into CK. At

45 °C and pH 3.5, the ginsenosides Rb1 and the enzyme exhibited the highest level of activity. The pathway Rb1 → Rd → F2 → CK converted roughly 84.3% of the ginsenoside Rb1 to CK one day after the incubation [53]. Furthermore, it has been observed that recombinant β-glu enzymes found in *Terrabacter ginsenosidimutans* sp. [54] and *Lactobacillus brevis* [55] can convert Rb1 into CK.

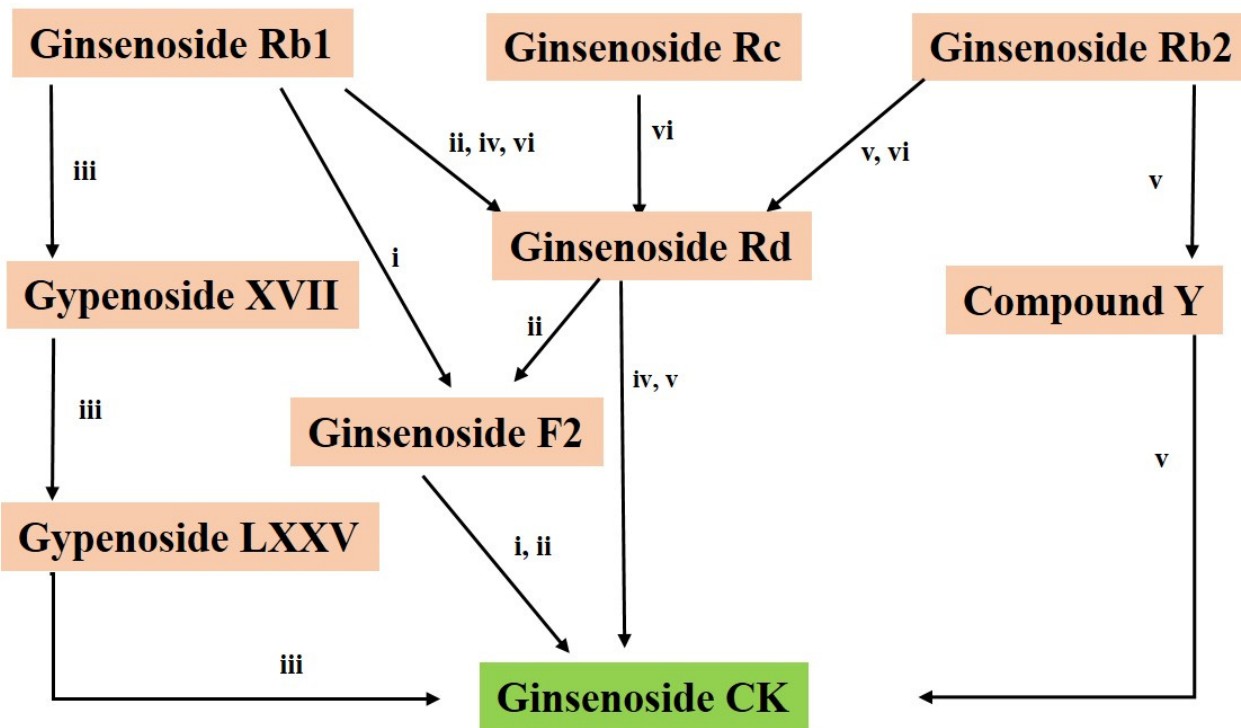

**Figure 1.** Biotransformation of major ginsenosides to CK via different pathways using enzymatic and microbial conversion methods. The numbers in the figure indicate the transformation-related enzymes. (i. β-Glucosidase from *Fusodobacterium K-60*, ii. β-Glucosidase from *Paecilomyces bainier*, iii. Recombinant β-glucosidase from *Terrabacter ginsenosidimutans*, iv. Recombinant β-glucosidase from *Microbacterium esteraromaticum*, v. β-Glycosidase from *Sulfolobus solfataricus*, vi. β-Glycosidase from *Pyrococcus furiosus*).

β-glycosidases are a substitute for β-glucosidases and are frequently employed in the hydrolysis of ginsenosides [56]. Particularly, PPD-type ginsenosides are hydrolyzed by β-glycosidases [57]. Noh et al. [58] documented synthesizing CK from ginseng root extract by employing β-glycosidases derived from *Sulfolobus solfataricus*. Two transformation pathways were described by them to turn Rb1, Rb2, Rc, or Rd into CK: (1) Rb1 or Rb2 → Rd → F2 → compound K, and (2) Rc → compound Mc → compound K. Despite the strong specificity of this approach, the ginsenoside to CK conversion rate was low. Thus, recombinant β-glycosidase derived from *Pyrococcus furiosus* [59] and *Microbacterium esteraromaticum* [60] has been created to convert significant ginsenosides into minor ginsenosides. *Pyrococcus furiosus* was highly productive in turning Rd into CK, yielding an 83% conversion rate [59].

However, there was inadequate stability of free β-Glu, which hindered circulation and recovery, unable to recycle β-Glu and snailase, self-digestion rate, and long reaction time. This limitation can be overcome by enzymatic immobilization technology, which will also make it easier to use β-Glu in commercial production β-Glu@Cu(PTA) biocomposite reached a 49.15% conversion rate of Rb1 to CK [50]. Green synthesis of Zn-BTC co-immobilized snailase and β-glucosidase (β-Glu) resulted in the formation of β-Glu&SN@Zn-BTC biocomposite, which reached the CK conversion rate of 53.5% in 48 h at pH 4.5. The CK concentration was 1.07 mg/mL, and 83% of all products were made up of CK [61]. Using Sna&β-Glu@H-Cu-BDC (large-sized snailase&β-glucosidase@hollow-

Cu-H2BDC) biocomposite for the synthesis of the CK. Cau et al. [62] reported that the total amount of CK was about 0.94 mg mL$^{-1}$, and the average conversion rate of CK was around 60.12% after growing the conversion system.

### 4.2. Biotransformation of CK by Human Gut Microbiota

Microbial transformation is generally regarded as a key technique for producing CK [63]. It involves using crude enzymes from *Fusarium sacchari* [64], *Lactobacillus paralimentarius* [65], *Microbacterium esteraromaticum* [60], *Caulobacter leidyia*, and *Acremonium strictum* [66]. For instance, Hasegawa et al. [67] thoroughly examined the metabolic conversion of CK from ginsenosides Rb1, Rb2, and Rc by gut flora. Ginsenosides Rb1, Rb2, and Rc were transformed into CK via anaerobic incubation with human gut microbiota. Among them, bacterial strains obtained from human intestinal feces, including *Bacteroides* sp., *Bifidobacterium* sp., and *Eubacterium* sp., successfully converted Rc into CK [68]. The composition of the gut microbiota determined the primary metabolic pathway used by intestinal bacteria to break down ginsenoside Rb1 into CK.

Compared to bacteria, fungi are easily cultured and can biotransform to replace human intestinal bacteria as a source of CK [64]. Zho et al. [69] used fungal biotransformation to efficiently manufacture CK from *Panax notoginseng* (PNG) saponins at a reasonable cost. The same group also showed that the fungus *Paecilomyces bainier* sp. 229 could efficiently convert PNG saponins into CK; this resulted in a substantially higher conversion rate of PNG to CK (82.6% vs. 35.4%) than before [70]. Furthermore, fungi that were produced from ginseng-cultivated soil, such as *Fusarium moniliforme* [71], *A. strictum* [72], *A. niger* [73], and *F. sacchari* [64], demonstrated good biotransformation of major ginsenosides into minor bioactive. Cumulative generation of bioactive CK from fermented black ginseng using a novel *Aspergillus niger KHNT-1* strain obtained from the Korean staple food kimchi [74]. *Leuconostoc* strains were also isolated from kimchi, which showed good conversion of PPD-type ginsenosides to CK [75]. Microorganisms have frequently been used in the biotransformation of major ginsenosides into minor bioactive. Furthermore, a synthetic biology approach has been utilized for transformation. PPD ginsenosides could be easily converted into CK by the metabolically modified yeast expressing the heterologous *UGTPg1* gene [76].

## 5. Mechanism of CK against Metabolic Diseases

The pathophysiology of metabolic disease is highly complex and is caused by multiple factors. Numerous studies have demonstrated the beneficial impacts of CK on metabolic disorders. After reviewing the research, we concluded that CK is a useful medicinal substance for treating osteoporosis, hyperlipidemia, obesity, hepatocyte steatosis, NAFLD, and diabetes and its consequences. Table 4 displays the pharmacological molecular pathways of CK in the treatment of metabolic disorders.

**Table 4.** Pharmacological action of CK in treating metabolic disorders and its underlying molecular processes.

| Disease | Experimental Models | Dosage Form | Doses of Administrations | Mechanism | Ref. |
|---------|---------------------|-------------|--------------------------|-----------|------|
| Obesity | C57BL/6J mice | Oral | 15, 30, 60 mg/kg | • Inhibits *TLR4/TRAF6/TAK1/NF-κB* activation in obese mice. <br> • Promotes *IRS1/PI3K/AKT* expression against obesity. | [13] |

**Table 4.** *Cont.*

| Disease | Experimental Models | Dosage Form | Doses of Administrations | Mechanism | Ref. |
|---|---|---|---|---|---|
| Obesity | Male C57BL/6J and ob/ob (B6/JGpt-*Lep*em1Cd25/Gpt) mice | i.p. injection | 20 mg/kg | • Initiates autophagy via the *AMPK/ULK1* pathway activation.<br>• Boosts autophagy and lipase activity to promote lipolysis.<br>• Promotes lipolysis via interacting with the *GR*. | [77] |
| | 3T3-L1 cell lines | Cell treatment | 0.05, 0.5, 5 μM | • Inhibits adipocyte-specific genes (*C/EBPα*, *leptin*, *aP2*, and *PPARγ*).<br>• Decreases angiogenic factors (*VEGF-A* and *FGF-2*) and MMPs (*MMP-2* and *MMP-9*).<br>• Enhances the mRNA expressions of angiogenic blockers (*TSP-1*, *TIMP-1*, and *TIMP-2*). | [19] |
| | 3T3-L1 cell lines | Cell treatment | 20, 50 μM | • Inhibits *MCP-1* and *TNF-α* in adipocytes.<br>• Promotes *IL-10* expression to alleviate obesity-induced inflammation and insulin resistance. | [78] |
| | 3T3-L1 cell lines | Cell treatment | 10–40 μM | • Activates the *AMPK* signaling pathway.<br>• Inhibits *ERK/P38* and *AKT* signaling pathways. | [79] |
| Diabetes | male ICR mice | Oral | 30 mg/kg/day | • Downregulates *PEPCK* and G6Pase expression in the liver. | [80] |
| | Male Wistar rats (200–250 g) | Oral | 30, 100, 300 mg/kg BW | • The levels of *InsR*, *IRS1*, *PI3Kp85*, *pAkt*, and *Glut4* in the skeletal muscle of diabetic rats may be enhanced by CK. | [81] |
| | MIN6 cell line | Cell treatment | 2–32 μM | • CK significantly stimulates insulin release by upregulating *GLUT2* expression. | [34] |
| | male C57BL/KsJ db/db mice | Oral | CK: Metformin 1:15 | • Insulin and plasma glucose levels were raised when CK and MET were combined. | [82] |
| DN | HFD (high-fat diet)/STZ (streptozotocin)-induced DN mice model | Intragastrically | 10, 20, 40 mg/kg/day | • Suppresses NADPH oxidase expression and blocks ROS-mediated *NLRP3* inflammasome and *NF-κB/p38* signaling pathway activation. | [83] |

<div align="center">**Table 4.** *Cont.*</div>

| Disease | Experimental Models | Dosage Form | Doses of Administrations | Mechanism | Ref. |
|---|---|---|---|---|---|
| DT | human tenocytes cell | Cell treatment | 3, 10 µM | • Inhibits high glucose-induced apoptosis, inflammation, and oxidative stress. <br>• Normalizes the *MMP-9*, *MMP-13*, and *TIMP-1* expressions. <br>• Boosts *PPARγ* and antioxidant enzymes. | [84] |
| OP | Raw264.7 cells <br> Balb/C female mice | Cell treatment, i.p. injection | 10 µM <br> 10 mg/kg | • Inhibits *RANKL*-induced osteoclast differentiation. <br>• Inhibits ROS production by triggering *NF-κB/p65* and oxidative stress in Raw264.7 cells. <br>• Inhibit bone resorption with macrophages generated from bone marrow. | [85] |
| | bone marrow mesenchymal stem cells <br> male Sprague Dawley (SD) rats | Cell treatment, i.p. injection | 2.5–40 µM <br> 10 µM | • Elevates *Runx2* and *β-catenin* to promote osteogenic differentiation via the Wnt/ β-catenin pathway. | [86] |
| OA | MC3T3E1 cell lines | Cell treatment | 0.01–10 µM | • Increases *ALP*, *Col-1*, and *Runx2* expression in preosteoblastic cells against osteoarthritis. | [8] |
| NAFLD | SD rats, <br> HSC-T6 cells | i.p. injection <br> Cell treatment | 3 mg/kg/day | • CK has anti-fibrotic and hepatoprotective effects. | [87] |
| | HepG2 cells | Cell treatment | 20 µM | • Suppresses *SREBP1c* and activates *PPAR-α.* | [30] |
| | HuH7 cells | Cell treatment | 1 µM | • Inhibits lipid droplet and triglyceride accumulation via upregulating the *AMPK/PPAR-α* signaling pathway. | [88] |
| HCC | HepG2 cells | Cell treatment | 0, 5, 10 µmol | • Enhances *P21* and *P27* expressions. <br>• Inhibits *cyclin D1*, *cyclin-dependent kinase 4*, and cell cycle progression to induce apoptosis. | [89] |

Abbreviations: DN: diabetes nephropathy, DT: diabetic tendinopathy, OP: osteoporosis, OA: osteoarthritis, NAFLD: non-alcoholic fatty liver disease, HCC: hepatocellular carcinoma, *TLR4*: Toll-like receptor 4, *TRAF6*: tumor necrosis factor receptor associated factor 6, *TAK1*: transforming growth factor-β-activated kinase 1, *NF-kB*: nuclear factor kappa B, *IRS1*: insulin receptor substrate 1, *PI3K*: Phosphoinositide 3-kinase, *AMPK*: AMP-activated protein kinase, *ULK1*: unc-51-like kinase 1, *GR*: glucocorticoid Receptor, *aP2*: activator protein 2, *PPARγ*: peroxisome proliferator-activated receptor gamma, *VEGF-A*: vascular endothelial growth factor A, *FGF-2*: fibroblast growth factor-2, *MMP*: matrix metalloproteinases, *TSP-1*: Thrombospondin 1, *TIMP*: tissue inhibitor of metalloproteinase, *MCP-1*: Monocyte chemo attractant protein 1, *TNF-α*: tumor necrosis factor-α, *PEPCK*: Phosphoenolpyruvate carboxykinase, *NADPH*: Nicotinamide adenine dinucleotide phosphate, *RANKL*: receptor activator of nuclear factor kappa-B ligand, *ROS*: reactive oxygen species, *RUNX2*: runt-related transcription factor 2, *ALP*: Alkaline Phosphatase, *Col-1*: Collagen 1.

### 5.1. Obesity

Obesity is a prevalent metabolic disease defined by adipocyte hypertrophy, which results from an imbalance between energy expenditure and food intake. Obesity has been

related to additional metabolic diseases such as insulin resistance, NAFLD, T2D, dyslipidemia, cardiovascular diseases, hypertension, and cancer [90]. Adipose tissue growth results in fat storage in pre-existing adipocytes and the transformation of preadipocytes into mature adipocytes, a process called adipogenesis [91]. Under adipogenic conditions, more amount of free fatty acids will be produced by hypertrophic adipocytes [92]. It shows that obesity is linked to a lipid metabolism issue; there is an indication that 43.2% of people with obesity have hyperlipidemia [93]. An important function of adipose tissue is to regulate energy metabolism [94]. It has been proposed that one of the key strategies for treating obesity is to activate brown adipose tissue (BAT) and produce browning in white adipose tissue (WAT) [95].

Compound K (CK) might be the potential therapeutic agent against obesity via several signaling pathways (Figure 2). A study showed that C5BL/6J mice consumed a high-fat diet to induce obesity, whereas the administration of CK (15, 30, and 60 mg/kg) might successfully enhance the resistance to insulin and glucose tolerance, downregulate *PPARγ* expression, inhibit *TLR4/TRAF6/TAK1/NF-κB* stimulation in obese mice, and lower macrophage M1-type inflammatory cytokine levels in serum and adipose tissue in a dose-dependent manner. Furthermore, CK increased *IRS1/PI3K/AKT* expression, which proved CK is an effective compound against obesity and early diabetes [13]. CK is a novel agonist of the glucocorticoid receptor (GR) used to treat obesity. In mice, CK was more effective than Orlistat in reducing blood lipids and weight [77]. CK treatment of 3T3-L1 adipocytes prevented lipid formation and the expression of genes particular to adipocytes (*C/EBPα*, *leptin*, *aP2*, and *PPARγ)*, decreased angiogenic factors (*VEGF-A* and *FGF-2*) and MMPs (*MMP-2 and MMP-9*), whereas enhanced the mRNA expressions of angiogenic blockers (*TSP-1*, *TIMP-1*, and *TIMP-2*) [19]. CK had strong inhibitory effects on the rise of *MCP-1* and *TNF-α* caused by the hypertrophic adipocyte supernatant. Additionally, it facilitated the expression of *IL-10*, prevented the induction of inflammatory macrophages, and enhanced the development of anti-inflammatory macrophages [78]. In early-stage adipogenesis, CK reduced the phosphorylation of protein kinase B (*AKT*), *p38*, and extracellular signal-regulated kinase [96]. Moreover, CK markedly elevated *AMPK* (AMP-activated protein kinase) and *ACC* (acetyl-CoA carboxylase) to suppress adipogenesis. In differentiated 3T3-L1 cells, the effect of CK on reducing *PPAR-γ* expression was restricted by *AMPK* pharmacological inhibition with dorsomorphin [79].

### 5.2. Diabetes and Related Complications

Diabetes mellitus (DM) is an emerging epidemic that can be linked to hereditary and environmental factors. Diabetes has complications that require treatment, including diabetic retinopathy, nephropathy, neuropathy, infertility, and cardiovascular disease. Type I diabetes, or T1D, and type II diabetes, or T2D, are the two primary subtypes of DM. An autoimmune condition called T1D kills beta cells in the pancreas and stops insulin from being released. On the other hand, T2D is characterized by high insulin levels and cell insulin resistance. According to epidemiological studies, there will be 629 million people with diabetes worldwide by 2045, up from a total of 425 million in 2017. Due to the potential harm that diabetes mellitus (DM) can inflict on an individual's quality of life, the condition needs to be controlled and managed as soon as possible [97]. To investigate anti-diabetic activity, ICR mice were fed CK (30 mg/kg/day) for 4 weeks. Phosphoenolpyruvate carboxykinase (*PEPCK*) and glucose-6-phosphatase (*G6Pase*), two glucose-producing genes, were downregulated after CK treatment. The results showed that CK can reduce blood sugar levels and increase insulin sensitivity in type 2 diabetes caused by a high-fat diet and fasting. This is achieved by suppressing the expression of *PEPCK* and *G6Pase* in the liver [80]. Jiang et al. studied CK (30, 100, and 300 mg/kg/BW) on male Wistar rats to improve insulin sensitivity. They found that CK may improve insulin resistance and hyperglycemia in diabetic rats. Additionally, studies indicated that CK could increase the expression of *Glut4*, *PI3Kp85*, *InsR*, *IRS1*, and *pAkt* in the skeletal muscle of diabetic rats. These findings suggest that increased insulin sensitivity, which is directly linked to

the *PI3K/Akt* signaling pathway, mediates the hypoglycemic action of CK [81]. Another study reveals that CK has strong stimulatory effects on insulin production in MIN6 cells by upregulating *GLUT2* expression (Figure 3) [34].

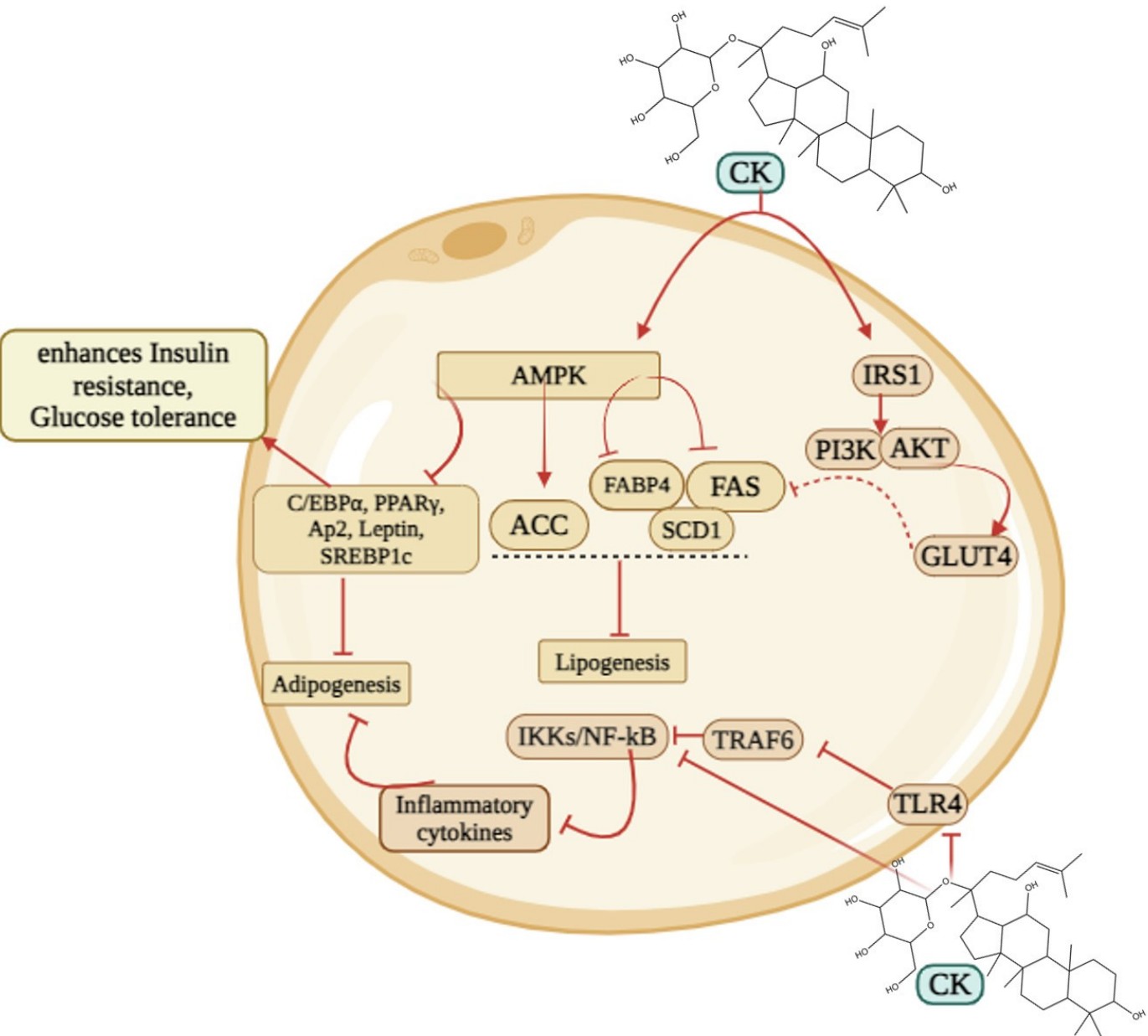

**Figure 2.** The pathway involved in obesity. CK inhibits adipogenesis via *AMPK/PPARγ/CEBPα* and lipogenesis via *AMPK/ACC/FAS* pathway. Additionally, CK increases *IRS1/PI3K/AKT* expression against obesity. Furthermore, CK triggers *TLR4/TRAF6/TAK1/NF-κB* to minimize adipose tissue.

Despite extensive research on the antidiabetic effects of CK, Song et al. studied CK on diabetic nephropathy (DN). DN mice models induced by HFD (high-fat diet) and STZ (streptozotocin) were administered CK intragastrically. The results demonstrated that CK dramatically reduced the growth of the glomerular mesangial matrix and considerably decreased the increased fasting blood glucose, serum creatinine, blood urea nitrogen, and 24 h urine protein of the DN mice [83]. Additionally, it was observed that the expression of *G6Pase* and *PEPCK* in the liver and HepG2 was suppressed by CK. In the meantime, *AMPK* activation was markedly boosted upon CK administration, but *FOXO1*, *HNF-4α*, and *PGC-1α* expressions were significantly decreased [98]. It has been noted that individuals

with diabetes with tendinopathy have a higher apoptotic tendency. Tendinopathy is a chronic illness that affects the tendons and causes a great deal of discomfort. It has a major negative influence on quality of life [99]. However, CK can effectively reduce the MMP system, inflammation, tenocyte apoptosis, and oxidative stress under hyperglycemia [84].

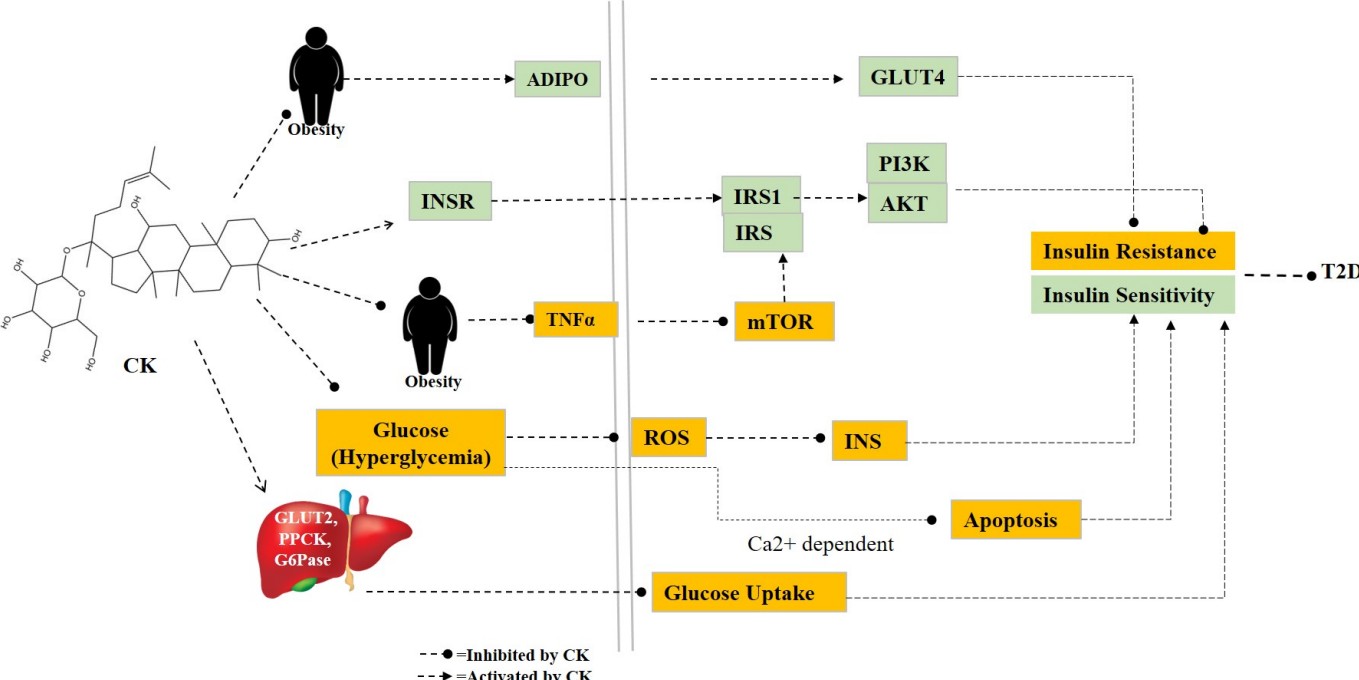

**Figure 3.** The pathway involves T2D. CK triggers insulin resistance and increases insulin sensitivity to inhibit T2D by regulating several genes. Abbreviations—CK: compound K, ADIPO: Adiponectin, INSR: insulin receptor, TNFα: tumor necrosis factor, GLUT: glucose transporter, PPCK: Phospho-enolpyruvate carboxylase kinase, G6Pase: Glucose-6-phosphatase, IRS: insulin receptor substrate, mTOR: mammalian target of rapamycin, ROS: reactive oxygen species, PI3K: Phosphatidylinositol 3-kinase, T2D: type 2 Diabetes.

*5.3. Osteoporosis*

Osteoporosis (OP) represents an alarming clinical condition that typically manifests as rapid bone loss during menopause. It increases the possibility of a brittle fracture, which puts a great deal of strain on society. A growing number of people are experiencing OP as society ages. Hip fractures are expected to become more common worldwide by 2050 as population demographics change. OP occurs when the bone resorption (osteoclast) rate is greater than the bone formation (osteoblast) rate, leading to lowered bone density [100]. Osteoblasts are bone-decomposed cells that play a crucial role in maintaining bone home-ostasis. During postmenopausal osteoporosis, the receptor activator for nuclear factor-κB ligand (*RANKL*) increases osteoclastogenesis, which results in bone loss [101]. Nowadays, many medications, including calcitonin and bisphosphonates, are used to treat and prevent OP. On the other hand, documented accounts of the adverse effects of these medications in medical settings exist. As a result, to lower fracture rates and enhance patient quality of life, improved therapies for OP must be investigated [102].

Herbal supplementations have been extensively researched as potential sources for drug development because of their lesser toxicities. As an herbal product, CK showed an anti-osteoporotic effect by suppressing RANKL-induced osteoclast differentiation and ROS production by triggering the *NF-κB/P65* signaling pathway [85]. Additionally, CK elevated the *Runx2* (master transcription factor) and *β-catenin* to promote osteogenic differentiation via the *Wnt/β-catenin* pathway [86]. Furthermore, CK has a preventive effect on osteoarthritis via upregulating *ALP*, *Col-1*, and *Runx2* in pre-osteoblastic MC3T3-E1 cell

lines [8]. Osteoprotegerin (OPG) on AA-FLS may be increased in vitro by CK, which can also lower RANKL levels and stop bone deterioration in AA rats [103]. Furthermore, pretreatment with CK may prevent human CD4+ monocytes and murine RAW264.7 cells from proliferating into TRAP+ osteoclast-like cells in response to soluble RANKL (sRANKL) in a dose-dependent way. Moreover, CK inhibited the nuclear transcription factor of activated T cells (*NFATc1*) and *RANK*-associated *NF-κB* pathways in osteoclast progenitors. These suggest that GCK blocked osteoclastogenesis caused by *RANKL* via two different pathways [33]. Based on all of these data, CK appears to be a promising treatment for preventing dietary induction of OP (Figure 4).

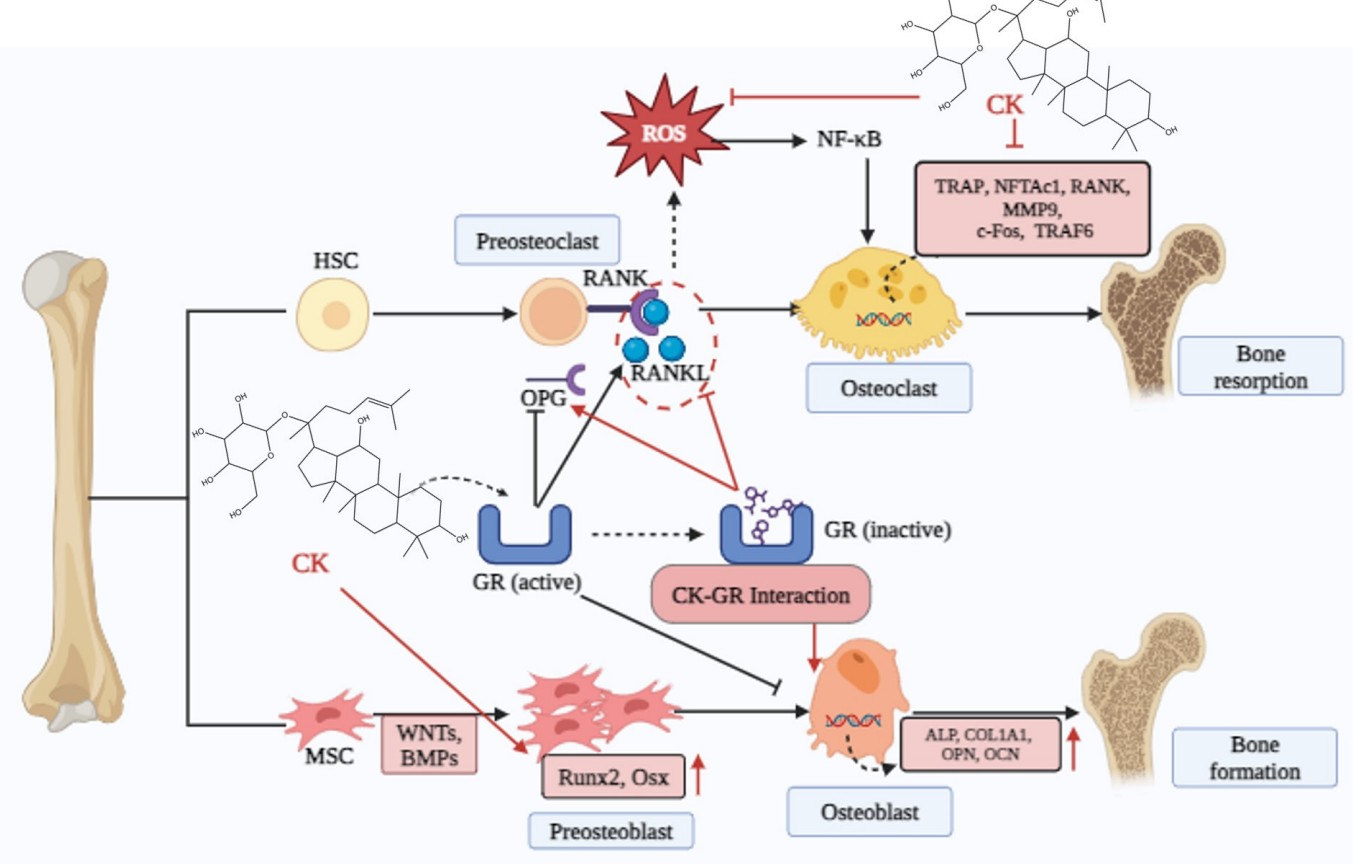

**Figure 4.** Compound K is effective in inhibiting osteoclast by interacting with glucocorticoid receptors, and this interaction triggers *RANKL,* which attaches to *RANK* receptor to produce osteoclast. CK also inhibits ROS and *TRAP, NFTAC1, MMPs, C-FOX*, and *TRAF6* to minimize bone resorption. On the other hand, CK increases the expression of preosteoblastic (*Runx2, Osx*) and Osteoblastic genes (*ALP, COL1A1, OPN, OCN*) that increase bone formation. Abbreviations—HSC: hematopoietic stem cell, MSC: mesenchymal stem cell, OPG: osteoprotegerin, GR: glucocorticoid receptor, RANKL: receptor activator of nuclear factor kappa-B ligand, Runx2: runt-related transcription factor 2, Osx: Osterix, ALP: alkaline phosphatase, COL1A1: collagen type 1, Alpha 1, OPN: Osteopontin, OCN: Osteocalcin, TRAP: Triiodothyronin receptor auxiliary protein, MMP_9: matrix metallopeptidase-9, TRAF6: TNF receptor-associated factor 6.

### 5.4. Non-Alcoholic Fatty Liver Disease (NAFLD)

Non-alcoholic fatty liver disease (NAFLD), also known as hepatic steatosis, or the formation of triglyceride in the liver, is not induced by consuming alcohol [104]. NAFLD is a broad term for a range of pathologies that include non-alcoholic fatty liver (NAFL), which is the first stage of NAFLD, non-alcoholic steatohepatitis (NASH), which is defined by the beginning of inflammation brought on by lipotoxicity, and severe NASH symptoms

that include fibrosis. According to reports, the overall incidence of NAFLD might reach 15% to 18% among Asian nations and 30% in Western nations. Obesity is closely linked to this fatty liver disease. The prevalence of NAFLD is predicted to increase globally in light of the present obesity pandemic [87]. A growing amount of clinical data indicates that NAFLD is a major risk factor for the emergence of liver cirrhosis, liver fibrosis, and liver cancer [105]. However, it is still unclear how NAFLD develops and how it leads to fibrosis and chronic liver disease. The "two-hit" theory first came forward in 1998: First, there is an initial metabolic change that results in insulin resistance, hyperglycemia, and hepatocyte triglyceride formation, which causes hepatic steatosis. The second hit causes the injury to worsen and progress, leading to cirrhosis, inflammation, fibrosis, and steatohepatitis [106]. On the other hand, the "multiple parallel hits" theory postulates that several parallel factors, including adipokines and cytokines secreted abnormally, dysfunctional mitochondria, stress to the endoplasmic reticulum, gut endogenous endotoxin, metabolism of lipids, lipotoxicity, oxidative stress, and genetic susceptibility cause NAFLD [107]. Still, the FDA has not approved any particular medications for NAFLD. Medicines such as atorvastatin calcium tablets and fenofibrate are commonly used to manage blood lipid levels [108]. However, using lipid-lowering medications can lead to certain negative side effects [109]. Therefore, finding novel medications that treat NAFLD with great efficacy and few adverse effects is urgent.

Various research has shown a significant potency of CK against NAFLD (Figure 5). Chen et al. demonstrated that CK is beneficial for treating NAFLD via hepato-protective and anti-fibrotic effects [87]. Another study depicted that CK activates AMPK and increases ACC and mononyl CoA levels in the AMPK signaling pathway to stimulate fatty acid oxidation. CK can suppress triglyceride accumulation in the liver by inhibiting lipogenic markers such as *SREBP1c*, *SDC1*, and *FAS* and enhancing lipolytic markers including *PPAR-α* and *CD36* [30]. *AMPK* also inhibits the formation of free fatty acids by reducing TG hydrolysis via direct phosphorylation and inactivation of hormone-sensitive lipase. Additionally, the activation of *AMPK* is related to elevated expression of *PPAR-α* and subsequent decrease of *SREBP1c* and *PPAR-γ* activity in adipocytes and hepatocytes. Moreover, CK might be directly mediating its beneficial impacts by activating *AMPK* [88]. To minimize the toxicity of CK, Yue et al. [110] developed a natural nano-CK that acts as an *mTOR* inhibitor to change lipid metabolism. In steatosis hepatocytes, nano-CK can alleviate lipotoxicity and restore lipid homeostasis by encouraging lipid export and blocking DNL and lipid absorption, all of which create a feedback loop regulated by *mTOR*. Furthermore, CK has a definite hepatoprotective impact on sodium valproate-induced hepatotoxicity, as shown by Zhou et al. [111]. These beneficial effects were mediated by reducing oxidative stress via the suppression of lipid peroxidation and the upregulation of the protective antioxidant system, controlling the peroxisome pathway via the downregulation of soluble epoxide hydrolase, and controlling iron homeostasis via the upregulation of hepcidin. In the case of hepatocellular carcinoma, CK caused a G0/G1 phase arrest, blocked cell cycle progression, and induced apoptosis via the upregulation of *p21Cip1* and *p27Kip1* and downregulation of *cyclin D1* and *cyclin-dependent kinase 4* in HepG2 cells. This was accomplished by the mitochondrial system via a modulation of the ratio of *Bcl-2* to *Bax* [89].

Thus, as monotherapy, in conjunction with other medications, or nanoformulation of CK may prove to be a promising therapeutic for alleviating hepatic problems associated with other metabolic diseases or diminishing fatty liver.

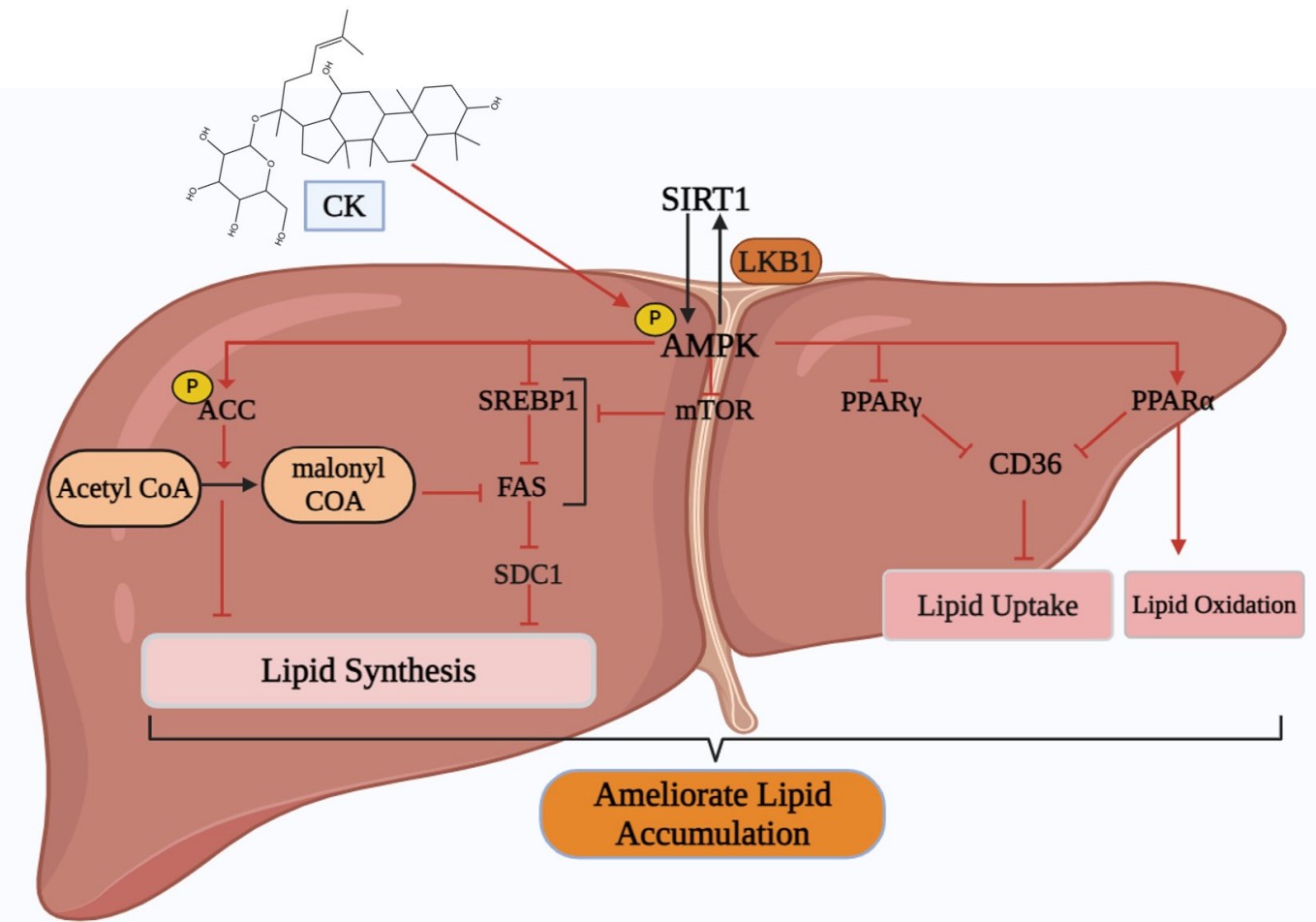

**Figure 5.** Activity of CK on the NAFLD pathway. CK increases the levels of the *AMPK/SIRT1* pathway that increases *ACC* and inhibits the *SREBP1/FAS/SDC1* pathway to minimize lipid synthesis. CK also uses *PPARα* to produce lipid oxidation. Abbreviations: SIRT1: Sirtuin 1, LKB1: Liver kinase B1, AMPK: AMP-activated protein kinase, mTOR: mammalian target of rapamycin, SREBP1: Sterol regulatory element-binding protein 1, FAS: fas cell surface death receptor, ACC: acetyl CoA carboxylase, SDC1: Syndecan 1, PPAR: Peroxisome proliferator-activated receptor, CD36: cluster of differentiation.

## 6. Synthesis of CK Analogues and Their Pharmacological Activity

Numerous biological actions of CK have been reported in the above sections. However, its use in medicine is limited by the lower membrane permeability, poor bioavailability, and water solubility of CK after oral administration, which is thought to be a limiting factor. Researchers are concerned with increasing the intestinal absorption of CK by modifying the structure. In the early stages of treating asthma, CK was found to exhibit strong action against IgE. In 2019, Ren et al. [112] reported couples of CK analogues were synthesized via straight forwarded methods. The produced compounds were assessed on the anti-IgE activities utilizing an in vivo airway hyper responsiveness experiment and an ovalbumin-induced asthmatic mouse model. They found that compounds T1, T2, T3, T8, and T12 (the analogues of CK) showed either superior or comparable anti-asthmatic effects compared to CK. Furthermore, Huang et al. [113] reported six derivatives of CK; among them, structures 1 and 2 were highly potent to activate the *LXRα* (Liver X Receptor *α*) expression and showed lower toxicity than CK. They also demonstrated that structures 1, 2, and 4 enhanced the expression of *ABCA1* (ATP-binding cassette transporter) mRNA levels. It has been documented that CK preferentially accumulates in the liver, where it converts to fatty acid esters.

Because the ester of CK was not removed by bile acid-like CK, it remained in the liver for a longer time. CK-octyl ester showed moderate detoxification and showed anti-liver cancer activity in murine-H22 cells and in vivo [114]. In addition, the novel ester prodrugs of CK (CK-butyl, and CK-octyl) have shown higher bioavailability due to their highly lipophilic properties than the CK. Furthermore, the findings indicated that the permeability coefficient of CK was lower than the esters [115]. Another CK-derived M1 and its analogues 1c, 2c, and 3c were documented by Li et al. as antitumor agents [116]. The findings suggested that M1 derivatives 2c and 3c had more potential to suppress the growth of triple-positive breast cancer and have opened up a new avenue for the production of possible anti-breast cancer medications. Different drug delivery technologies, including liposomes, nanoparticles, and micelles, have been created to increase the efficacy of the active components. For example, Yang et al. demonstrated that CK-loaded liposomes increase water solubility and cellular uptake than CK [117]. Mesoporous silica nanoparticles of CK showed the best biocompatibility in HaCat cells and anti-inflammatory activity compared to standard CK [118]. A54 peptide has recently been utilized to create CK-loaded micelles (APD-CK) intended for liver targeting. The findings demonstrated that APD-CK micelles increased the absorption of micelles by cells and stimulated cell death in HepG2 and Huh-7 cells in vitro [119]. These findings establish a groundwork for further alterations of CK and apply the new structures to metabolic diseases.

## 7. Discussion and Future Perspective

According to reviewed databases involving numerous studies, we summarized that CK is mostly associated with metabolic disorders, such as obesity, NAFLD, OP, diabetes mellitus, its complications, and the related pathways. It demonstrates the variety of ways in which CK can contribute to metabolic diseases such as enhancing IR, inhibiting glucose uptake, boosting glucose tolerance and insulin sensitivity, inhibiting bone resorption, increasing bone formation, triggering lipid synthesis, lipid uptake, boosting lipid oxidation, and blocking the inflammatory cytokines.

In this study, we demonstrated, firstly, that CK enhances AMPK and inhibits adipogenic genes (*C/EBPα*, *PPARγ*, *Ap2*, *leptin*, and *SREBP1c)* and lipogenic genes (*FAS*, *FABP4*, and *SCD1*). CK also inhibits the *IKKs/NF-kB* pathway to trigger obesity mediated-inflammation. Furthermore, CK increases the expression of *Glut4*, *PI3Kp85*, *InsR*, *IRS1*, and *pAkt* to block the IR and increases *GLUT2/PPCK/G6Phase* pathway to reduce gluconeogenesis, inhibits ROS/INS and apoptosis to increase insulin sensitivity. Moreover, inactivated GR inhibits osteoclast differentiations and bone resorption and increases osteoblastic differentiation. Osteoclast production is inhibited, and osteoblast development is promoted by the stimulation of the activity of genes associated with osteogenesis, such as *Runx2*, *osterix* and the suppression of the expression of genes related to osteoclasts, such as C-Fos and NFAC1. Meanwhile, CK is a novel agonist of the GR to treat obesity and OP. Lastly, CK ameliorates lipid accumulation by activating the *AMPK/SIRT1* pathway in the case of NAFLD. Research on CK in the future may focus on atherosclerosis, cardiovascular disease, fatty liver, hyperlipidemia, and other conditions.

Although numerous studies have established the toxicity of CK, our review revealed that the toxicity of CK is mostly dependent on the dosage and timing of administrations, and the sex of subjects also affects hepatotoxicity. The two animal species that should be included in medication safety test screenings are rodents and non-rodents (typically dogs), according to the most recent clinical guidelines [120]. Oral administration of CK in mice (10 g/kg) and rats (8 g/kg) did not exhibit any toxicity or death. Another study was performed to evaluate 26-week toxicity (food consumption, body weight, hematological parameters, and histopathology of rats) at different dosages ranging from 13 to 120 mg/kg of CK. The results suggested that 40 and 120 mg/kg doses had no observed adverse effect levels for females and males, respectively. However, in the male test group receiving 120 mg/kg, there was a brief fall in body weight, fur loss, decreased activity, and a shortage of energy. Beagle dogs were administered 4, 12, and 36 mg/kg oral doses for 26 weeks and

did not exhibit any visible toxicity in the 4–12 mg/kg groups. Groups treated with 36 mg/kg showed decreased body weight, elevated plasma enzymes, and nephrotoxicity [47].

Studies showed that the dose of CK administrations did not exceed 100 mg/mL body weight ranging from for treating metabolic diseases. Additionally, in the cell lines treatments, the range of administration was 0.1–64 μg/mL showed more than 80% cell viability. However, several cells showed cell cytotoxicity in different concentrations. Thus, it can be suggested that it is difficult for CK to cause bio-toxicity at normal doses. The development of bio-toxicity is more closely linked to an increase in CK dose than it is to an extension in administration time. Generally, doses of CK used to treat metabolic disorders in mice and rats are less than 100 mg/kg. However, doses up to 120 mg/kg in mice and rats can cause hepatotoxicity and nephrotoxicity [38]. It has been proposed that additional assessments are required to confirm the safety of administering CK to humans.

It is well recognized that the potential for the treatment of CK is usually limited because of poor water solubility, bioavailability, and membrane permeability [121]. When it comes to medications that are poorly soluble in water, co-crystals (a supernatural delivery system) can increase the bioavailability of drugs without altering their pharmacological action. The goal of cocrystallization research has been to maximize the physicochemical characteristics of pharmaceuticals, including their mechanical qualities, stability, solubility, permeability, and bioavailability. Enhanced solubility of cocrystals has been linked to increased gastrointestinal absorption of cocrystals in animal experiments. Enhancing oral bioavailability via the cocrystallization of medicines with suitable conformers is a potential strategy. Interests and Suglat are co-crystal drugs available in the market that improved bioavailability than the parent drugs [122]. Furthermore, some studies have attempted to create more CK or change the structure via different methods. For example, combining CK with targeted carriers improved water solubility, which increased bioavailability and low cytotoxicity [121,123].

Thus, it is recommended that co-crystals of CK undergo research as an antimetabolite to increase oral bioavailability. Another recommendation is to change the chemical structure of CK or modify its dosage forms such that it dissolves better in intestinal fluids and has a higher oral bioavailability. Research on anti-metabolic disorders and other CK-related pharmacological effects should focus on comparing and examining the safety and pharmacological effects of injectable and oral CK metabolites in vivo. In addition, researchers are interested more in CK analogues due to their less cytotoxicity, more bioavailability, better membrane permeability, and higher efficacy compared to CK in various diseases. CK analogues could be drug candidates because of their physiochemical properties and pharmacological action.

The review sought to present up-to-date data on the safety, pharmacokinetics, and health-promoting properties of CK and its analogues in the management and prevention of disease. It is commonly recognized that CK has several health advantages and is more permeable than its parent saponins. Despite being more bioavailable than other ginsenosides, the therapeutic application of CK is limited by a few issues. It has been demonstrated that using CK derivatives as nanocarriers improves their permeability, solubility, and efflux, as well as their capacity to promote health. To sum up, there are few pharmacokinetic investigations on monomer CK and the experimental and clinical safety data associated with it. To evaluate the effectiveness and safety of CK and its derivatives, particularly in clinical trials, more research is necessary.

**Author Contributions:** Methodology, M.N.M.; writing—original draft, M.N.M.; review and editing, R.A., M.R.K. and S.I.; supervision, S.C.K. and D.C.Y. All authors have read and agreed to the published version of the manuscript.

**Funding:** This research received no external funding.

**Data Availability Statement:** Data are contained within the article.

**Conflicts of Interest:** The authors declare no conflicts of interest.

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
