# Peer review of "Bioconversion, Pharmacokinetics, and Therapeutic Mechanisms of Ginsenoside Compound K and Its Analogues for Treating Metabolic Diseases"

_cimb, doi:10.3390/cimb46030148_

Round 1

Reviewer 1 Report

Comments and Suggestions for Authors

The submitted manuscript concerns the data on ginsenoside compound K. It is quite interesting. The Authors consider the pharmacokinetics of ginsenosides which are metabolized to CK. However, what about the pharmacokinetic fate of CK? Therefore, some statements such as in L. 305 are far-fetched. There is too little data to state that CK might be a medicine.

More details like in the ref. 25. In L. 90-92 the statement must be rearranged. The ref. 25 provides other data than in this statement. There are no biological studies in this report. Please develop e.g. “Metabolism pathway of CK have been proposed, including prototype oxidation, deglycosylation, deglycosylation with sequential oxidation and dehydrogenation, deglycosylation with sequential glucuronidation.” Such information was missed in the submitted manuscript.

What is the source of the data provided in the supplementary material? Please move it to the main text and provide a reference.

A review of the pharmacological activity of CK was published: Liu T, Zhu L, Wang L. A narrative review of the pharmacology of ginsenoside compound K. Ann Transl Med. 2022 Feb;10(4):234. doi: 10.21037/atm-22-501

Minor comments:

L. 129 should be “ginsenoside”

Please check the data in Table 2. Effects should be shortly described, not necessarily in the sentences. Please check the spelling, e.g. “toxicitry”

Figure 1 should be enlarged to show better structures. Could you provide the names of enzymes that catalyze the transformation reaction? In some cases, other enzymes than glucosidase can participate in the process, e.g. it seems in the case of Rg3 to CK.

There are a lot of abbreviations, which development was missed or it is difficult to find, e.g. L. 210 “Mc”, L. 245 CNS. It is worth to provide a list of abbreviations.

L. 231 “Please check “Lactobassillus”

L. 248-249 The statement must be checked. No lower cases in the sentence.

Table 3. Pharmacological effects concern in vivo models. For this reason, in vitro and in vivo models should be separated. All abbreviations must be developed in the footnotes.

Figure 3. The structure of the compound must be enlarged. All abbreviations must be developed.

Figure 4. All abbreviations must be developed. The names of genes must be in italics.

L. 375 Please do not use abbreviations at the beginning of the sentence, e.g. “CK”

Section 6 must be seriously rearranged or removed. There is a weak relationship with CK studies. p53 must be written.

L. 539-551 This paragraph must be rearranged. Please do not use “1)…2)…3)…4)…” Suggested native speaker’s advice.

L. 559 36 g/kg at the beginning of the sentence. Please change it.

L. 583-585 Please rearrange or provide more pharmacokinetic data of CK itself.

Author Response

Dear Reviewer,

Thanks for your valuable comments. We revised our manuscript to meet the journal standard. Now, we think, the manuscript might be easily understandable to the reader.

Here I have attached the response letter to the reviewer.

Reviewer 2 Report

Comments and Suggestions for Authors

The authors have provided a detailed account of the biological activities of CK, its pharmacological actions, and the signaling pathways it modulates. The review also discusses the bioavailability, toxicity, and pharmacology of CK, setting a foundation for future clinical studies. While the manuscript is informative and covers a significant amount of research, there are several areas that require attention to enhance the clarity, depth, and scientific rigor of the study.

The discussion section could be more focused on synthesizing the findings and presenting a critical analysis of the current state of research, including gaps and future directions.

The authors mention the poor water solubility and bioavailability of CK as major limitations for its therapeutic use. However, the discussion on strategies to overcome these limitations, such as the use of co-crystals and chemical modification, is brief. A more detailed exploration of these strategies, including specific examples and their outcomes, would strengthen the manuscript.

The review touches upon the comparison between CK and its analogues in terms of cytotoxicity, bioavailability, and efficacy. However, this comparison lacks depth. The authors should provide a more detailed comparative analysis, including quantitative data where available, to substantiate the claims made regarding the superiority of CK analogues.

The manuscript discusses the dosage-dependent toxicity of CK but does not provide a comprehensive analysis of the dose-response relationship. A more detailed discussion on the safe dosage range, the mechanisms underlying toxicity, and how these might vary between different animal models and human subjects would be valuable.

While the review mentions pharmacokinetics studies, there is a lack of detailed data on the absorption, distribution, metabolism, and excretion (ADME) of CK and its analogues. The authors should include more pharmacokinetic parameters to provide a complete picture of how CK behaves in biological systems.

The manuscript would benefit from a discussion on the translational potential of the findings. This includes any existing clinical trials, the challenges in translating preclinical findings to the clinic, and the potential for CK and its analogues to be developed into marketable drugs.

Comments on the Quality of English Language

The English quality of the manuscript is fine, requiring minor editing

Author Response

(The authors gave the same response as above.)

Round 2

Reviewer 2 Report

Comments and Suggestions for Authors

The authors improved the manuscript following my requirements. I consider the manuscript suitable to be published in this form.